# Endoscopic Management of Large Non-Pedunculated Colorectal Polyps

**DOI:** 10.3390/cancers15153805

**Published:** 2023-07-27

**Authors:** Oliver Cronin, Michael J. Bourke

**Affiliations:** 1Department of Gastroenterology and Hepatology, Westmead Hospital, Sydney, NSW 2145, Australia; 2Westmead Clinical School, University of Sydney, Sydney, NSW 2145, Australia

**Keywords:** colonoscopy, polyp, polypectomy, colorectal cancer, endoscopic mucosal resection, endoscopic submucosal dissection

## Abstract

**Simple Summary:**

Endoscopic resection (ER) of large non-pedunculated colorectal polyps ≥ 20 mm (LNPCPs) is safe, effective and the preferred treatment compared to surgery. Predicted histopathology of an LNPCP based on size, morphology, granularity, pit pattern and location in the colo-rectum is essential when deciding upon resection technique. Post resection defect inspection and adjuvant techniques, such as thermal ablation of the margin, have been demonstrated to reduce recurrence rates. Follow-up surveillance colonoscopy can accurately identify recurrence. Endoscopic treatment of recurrence is effective.

**Abstract:**

Large non-pedunculated colorectal polyps ≥20 mm (LNPCPs) comprise approximately 1% of all colorectal polyps. LNPCPs more commonly contain high-grade dysplasia, covert and overt cancer. These lesions can be resected using several means, including conventional endoscopic mucosal resection (EMR), cold-snare EMR (C-EMR) and endoscopic submucosal dissection (ESD). This review aimed to provide a comprehensive, critical and objective analysis of ER techniques. Evidence-based, selective resection algorithms should be used when choosing the most appropriate technique to ensure the safe and effective removal of LNPCPs. Due to its enhanced safety and comparable efficacy, there has been a paradigm shift towards cold-snare polypectomy (CSP) for the removal of small polyps (<10 mm). This technique is now being applied to the management of LNPCPs; however, further research is required to define the optimal LNPCP subtypes to target and the viable upper size limit. Adjuvant techniques, such as thermal ablation of the resection margin, significantly reduce recurrence risk. Bleeding risk can be mitigated using through-the-scope clips to close defects in the right colon. Endoscopic surveillance is important to detect recurrence and synchronous lesions. Recurrence can be readily managed using an endoscopic approach.

## 1. Introduction

Colorectal cancer (CRC) is the third most commonly diagnosed malignancy and the second most frequent cause of cancer-related death [1,2]. The majority of CRCs arise via the stepwise acquisition of molecular abnormalities in the adenoma–carcinoma and serrated pathways [3,4,5]. This creates the opportunity for intervention to remove premalignant polyps. Endoscopic resection (ER) of pre-malignant polyps has been shown to reduce the incidence of CRC [6,7,8]. Moreover, screening colonoscopy and polypectomy have been shown to reduce the risk of death from CRC at 10 years (risk ratio 0.82, 95% confidence interval (CI) 0.70–0.93) [9]. In a large study (*n* = 2602) with follow-up over 23 years, a 53% reduction (relative risk (RR) 0.47; 95% CI 0.26–0.80) in mortality was demonstrated in those who had undergone polypectomy [7].

The majority (90%) of colorectal polyps are <10 mm in size, do not contain advanced pathology and can be removed either en bloc or piecemeal using cold-snare polypectomy (CSP) [10,11,12]. Large non-pedunculated colorectal polyps ≥ 20 mm (LNPCPs) comprise ~1% of all colorectal polyps. These lesions have varied risk of overt and covert submucosal invasive cancer (SMIC), and therefore require a detailed, methodical optical assessment before deciding on the most suitable resection technique [13,14,15,16]. This algorithm needs to account for LNPCP size, morphology, location and pit pattern in addition to any patient-specific factors, such as co-morbidities and anticoagulation or anti-platelet medications [17].

Consensus recommendations favour an endoscopic approach as first line for the resection of LNPCPs (based on high-quality evidence) [18,19]. Compared to surgical resection, EMR has been demonstrated to have reduced morbidity and mortality and lower healthcare costs [20,21].

ER can be divided into three discrete phases: pre-resection, resection and post-resection. The technical success of ER requires a methodical, collaborative approach, ideally at a centre with access to the complete range of ER techniques, including conventional endoscopic mucosal resection (EMR), cold-snare EMR (C-EMR) and endoscopic submucosal dissection (ESD) (Figure 1). This review aimed to provide a comprehensive, critical and objective analysis of ER techniques. Herein, we outline an evidence-based approach to the ER of colorectal polyps.

## 2. Pre-Resection

Planning is essential to ensure technical success. The planning phase can be sub-divided into pre-procedure and intra-procedure.

Pre-procedural planning starts with patient assessment, accounting for frailty, functional status, co-morbidities and medications. Consent must include the risks and benefits of ER and a discussion around alternative modalities, such as surgery. Predicted lesion histopathology, including the risk of SMIC, should influence ER modality, and any related imaging should be reviewed. Pre-procedural planning also includes an in-room discussion with the endoscopy team to ensure that nursing and anaesthetic staff are aware of the various stages of the procedure, including any site-specific challenges, such as those seen with ileocaecal valve (ICV) lesions [17,22]. The pre-procedure discussion with the endoscopy team should also include the expected procedure time, any required medications, such as surgical antibiotic prophylaxis or local anaesthetic for anorectal junction (ARJ) lesions, and a check to ensure appropriate snares and ESD knives are available [23,24]. ER should only be performed using carbon dioxide insufflation [25]. Required ancillary devices should be in the room pre-procedure, including closure devices, such as through-the-scope clips, and those used to treat intra-procedural bleeding, such as haemostatic forceps.

Intra-procedural planning starts with patient positioning. The optimal patient position is to have the fluid pool opposite the lesion to maximise the effect of gravity on lesion elevation and achieve a clear working field during tissue resection or for management of any complications. Therefore, a supine or right lateral position may be required. Position of the colonoscope to align the lesion at a 6 o’clock position is essential. Dependent on location, a retroflexed position may improve access and optical assessment.

Thorough optical assessment is key. The risk of overt (optical features of SMIC present) or covert (optical features of SMIC absent) cancer can be predicted based on LNPCP size, location, morphology, granularity, and microvascular and surface pit patterns [13,26]. Several classification systems exist, including the Kudo pit pattern (KPP) and the Japan Narrow-Band Imaging Expert Team (JNET) classification [27,28]. An understanding of these systems is useful. A simple innovation to assist with familiarly and use is to place large posters of these classification systems in endoscopy rooms and reporting areas. Benign lesions have surface homogeneity with a regular pit and microvascular pattern (Figure 2). High grade dysplasia or cancer within a benign lesion appears as a demarcated area of disruption within this regular pattern (Figure 3). Such areas need to be very carefully examined to ensure the correct optical diagnosis and treatment strategy.

Traditionally, the accuracy of optical diagnosis for SMIC was evaluated across the entire LNPCP spectrum and was found to have suboptimal utility. Recently, optical assessment of flat (Paris 0-IIa) LNPCPs has been proven to be highly accurate [15]. In a large, prospective, single-centre cohort study (*n* = 1583), the sensitivity and specificity for predicting cancer in Paris 0-IIa LNPCPs was 91% and 96%, respectively. The likelihood that cancer would be missed in this study was 6 in 1000 cases. Optical diagnosis for SMIC in nodular lesions is less accurate (sensitivity 53%, specificity 94%, missed SMIC 6%) [15]. Excluding those lesions with overt SMIC, a large multicentre, prospective study (*n* = 2277) found that covert SMIC was associated with Paris 0-Is and Paris 0-IIa+Is morphology, non-granularity, size and distal location [13]. Supporting this, a large prospective cohort study (*n* = 3405) demonstrated that nodular rectal LNPCPs are more likely to contain SMIC than non-rectal colonic LNPCPs (15% vs. 6%, *p* < 0.001) [26].

## 3. Resection

In 2023, a selective resection algorithm should be employed when considering a therapeutic strategy for any colorectal polyp or neoplasm. This is based on optical diagnosis for predicted histology, lesion size, morphology, surface granularity and location in the colon.

### 3.1. Diminutive (<5 mm) and Small (5–9 mm) Colorectal Polyps

The overwhelming majority of colonic polyps are diminutive (<5 mm) or small (5–9 mm). CSP is safer and equi-efficacious compared to hot-snare polypectomy (HSP) for the removal of these colorectal polyps. The absence of electrocautery all but eliminates the risks of perforation and post-polypectomy bleeding [29,30]. Based on high quality data, en bloc or oligo-piecemeal CSP should be used to resect these polyps [18,19].

### 3.2. Medium (10–19 mm) Colorectal Polyps

There is a paradigm shift toward C-EMR given its superior safety profile. A large, prospective, multicentre cohort study (*n* = 286 lesions) comparing conventional EMR to C-EMR for 6–15 mm polyps favoured the use of C-EMR over EMR [31]. At present, US consensus guidelines recommend either EMR or C-EMR for resection of lesions 10–19 mm [18].

### 3.3. Large (>20 mm) Non-Pedunculated Colorectal Polyps

Conventional EMR is the mainstay for ER of LNPCPs due to its superior safety, efficacy and cost effectiveness compared to surgery and ESD [17,18,19,22]. High-quality studies over the past 10–15 years have lead to improvements in the safety and efficacy of EMR. These include the use of CO_2_ for insufflation; addition of chromo-injectate into the submucosal space; use of a systematic inject and sequential snare resection technique; removing a 2–3 mm margin of normal mucosa; water expansion of the defect to identify any residual adenoma; and recognition and management of significant DMI [25,32,33]. When all visible adenoma has been excised, thermal ablation of the margin should be completed by gently applying snare-tip soft coagulation (Effect 4, 80 W: ERBE Electromedizin, Tubingen, Germany), aiming for a 3–5 mm rim of ablated mucosa [34]. In a large, prospective cohort (*n* = 390) comparing conventional EMR without and with thermal ablation, recurrence rates reduced from 21.0% (37/176) to 5.2% (10/192), *p* < 0.001. No adverse events were attributed to margin thermal ablation. Since its inception, application of this adjuvant technique has improved. In a recent, larger multicentre cohort (*n* = 1049), recurrence rates at 6-month follow-up colonoscopy (SC1) were 1.4% (10/707) [35].

At present, given the paucity of data, conventional EMR is recommended over C-EMR for LNPCP resection. The safety profile of C-EMR is appealing for the piecemeal resection of Paris 0-IIa (flat, sessile) LNPCPs; however, the upper size limit that can be effectively removed using C-EMR without excessive burden of recurrence is unknown. Several ongoing large randomised controlled trials comparing EMR and C-EMR for non-serrated LNPCPs (clinicaltrials.gov identifier: NCT04138030; NCT04418843) aim to provide clarity on this issue. The next important RCT will compare C-EMR to C-EMR with thermal ablation of the margin (clinicaltrials.gov identifier NCT05041478).

In contrast to adenomatous LNPCPs, C-EMR is always the primary modality for ER of serrated LNPCPs, irrespective of size [36]. A large study (*n* = 562) of serrated lesions found no difference in technical success and recurrence rates between EMR and C-EMR groups; however, bleeding (0% vs. 5.1%) and significant deep mural injury (DMI) (0% vs. 2.8%) were more common in the EMR group.

### 3.4. Special Considerations

Site specific considerations and technique modifications may be needed for LNPCPs located at the ICV, appendiceal orifice, surgical anastomosis, or an ARJ or those which are circumferential [16,37,38,39].

The rectum should be regarded as a complex high-risk site, with distinct challenges compared to the colon. This is not due to its technical limitations, but due to its increased risk of covert SMIC [26,40]. Furthermore, the consequences of failed endoscopic cure include consideration of the most hazardous and complicated forms of colorectal surgery, including permanent ostomy formation [41]. Patients with rectal lesions removed using a low or ultra-low anterior resection have an increased risk of incontinence (12%) [42] and sexual dysfunction (20–46%) [43], and a 10–20% risk of permanent stoma [44,45]. Low anterior resection has a 30-day morbidity and mortality of 25% and 6%, respectively [46]. Postoperative complications have been associated with negative economic impact, increased morbidity, extended postoperative hospital stay, readmission, sepsis and death. ER is organ-sparing and minimally invasive, which enables avoiding wound infections as well as other postoperative complications after open surgery, which cause pain and suffering to patients [47].

In a large, multicentre observational study (*n* = 618), rectal LNPCPs were more likely to have nodular morphology (53% vs. 17%, *p* < 0.001) and contain cancer (15% vs. 6%, *p* < 0.001) compared to LNPCPs in the remainder of the colon [26]. Endoscopic en bloc resection for any LNPCP with a nodular component is critical with the aim of achieving an R0 (curative) resection. This requires meticulous planning.

ESD was developed as an ER technique for the curative treatment of early gastric cancer. ESD is now an established technique in the colo-rectum. It is typically performed with a generous submucosal injection, in a retroflexed position for improved scope stability, and using an improved more parallel angle of the cutting plane. Dissection is performed using an electrosurgical knife. Technique has improved over the past 10 years, aided by internal and external traction devices as well as techniques such as pocket creation ESD.

EMR and ESD are complementary techniques for resection of rectal LNPCPs. A selective resection algorithm (SRA) has demonstrated superior outcomes compared to a universal EMR algorithm (UEA). In a large study (*n* = 480) comparing an SRA to a UEA, LNPCPs underwent ESD if they had features suggestive of superficial overt SMIC (1000 μm, KPP V_I_) or covert SMIC (Paris 0-Is or a dominant nodule). All (*n* = 7, 100%) LNPCPs with SMIC amenable to R0 resection that underwent ESD were cured [16]. A rectum-specific SRA avoids the piecemeal resection of cancer.

Until recently, the management of covert SMIC discovered after piecemeal ER has been challenging. A recent observational study (*n* = 3372) identified 143 (4.2%) cases with covert SMIC post piecemeal resection [48]; 109 cases underwent surgical resection, and 62 (63%) cases had no residual cancer. All cases with residual intramucosal cancer (*n* = 24) could be identified by a R1 histological deep margin. Cases with poor differentiation and/or lymphovascular invasion had a high risk of lymph node metastases (12/33); there was a very low risk without these features (<1%, 0/35). The majority of patients with covert SMIC resected piecemeal had no residual malignancy. The risk of malignancy can be predicted by poor differentiation, lymphovascular invasion and an R1 deep margin.

Prevention of bleeding by prophylactic treatment of medium and large vessels with coagulating forceps is key. Bleeding stains the mucosa, impeding views, and leading to a higher risk of incomplete resection. Treatment of bleeding can char the mucosa, also obscuring views. Given its resource intensive, time consuming nature, this technique is best reserved for lesions with superficial overt SMIC or a high risk of covert SMIC. In clinical practice, this limits its use predominantly to the rectum [24].

Previously attempted LNPCPs are common and present a unique set of challenges. Due to the dense submucosal fibrosis, submucosal lift if often unsuccessful. A large observational study (*n* = 1292) demonstrated that with the use of auxiliary these lesions can be effectively resected by EMR. CAST was used in 73 (46.2%) cases. No recurrence (*n* = 0, 0%) was identified in any previously attempted LNPCPs that underwent margin thermal ablation, demonstrating that EMR is effective for resection of these lesions [49].

### 3.5. Complications

#### 3.5.1. Deep Mural Injury

Significant DMI (Deep Mural Injury Types III–V) was previously a feared intra-procedural complication, with a frequency of approximately 3% [40]. However, due to an improved understanding of risk factors, earlier recognition and advances in closure devices, such as through-the-scope clips, significant DMIs can now be successfully managed [40,50]. In a large, prospective cohort (*n* = 911), significant DMI was associated with attempted en bloc resection, advanced histopathology and transverse colon location [50]. In a large, prospective cohort (*n* = 3717), significant DMI occurred in 2.7% (101/3717) of EMR resections (median lesion size 35 mm, interquartile range 25–45 mm). Successful defect closure occurred in 97.0% (98/101) of cases. There were no differences found between DMI and non-DMI cases in terms of technical success or recurrence [40].

#### 3.5.2. Post-Procedural Bleeding

Prophylactic treatment of visible vessels within a defect post EMR has been previously investigated. In a multicentre RCT (*n* = 347, 55.3% proximal colonic lesions), prophylactic endoscopic coagulation of all visible vessels within the post-EMR defect did not reduce clinically significant post-EMR bleeding compared to no treatment (5.2% vs. 8.0%; *p* = 0.30) [51]. Post-resection defect closure for right-sided lesions using through-the-scope clips has been shown to reduce clinically significant post-EMR bleeding from 10.6% (12/113) to 3.4% (4/118), *p* = 0.031 [52].

Post-ER bleeding has a frequency of 6–7%, dependent on defect location and the selected ER modality. Bleeding typically does not require intervention, and these cases are managed conservatively in >50% of cases [53].

## 4. Post-Resection

### 4.1. Post-Operative Care

Post-resection instructions and communication with nursing staff, patients and their next-of-kin are important to ensure early recognition and management of any adverse events or complications. Recovery staff should receive a verbal handover and a written endoscopy report from the proceduralist, including any complexities or nuances of the case. Dependent on the procedure type, patients should remain fasting for at least 2 h or until they have been examined by the proceduralist. After clinical assessment, if the patient is well, they can commence a clear fluid diet.

The patient should receive a copy of their report. Dietary instructions should be highlighted and details of the best hospital contact should be clear, should the patient have any issues or questions overnight. Most patients can be discharged home the same day, but an endoscopy team member should contact the patient the following day for a telehealth assessment.

### 4.2. Surveillance

Guidelines recommend a follow-up surveillance colonoscopy 6 months post-ER [18,19,54]. Surveillance post-ER is essential to evaluate the previous resection site and to exclude synchronous lesions [55]. Co-existent advanced pathology (polyps > 10 mm or with a villous component or high-grade dysplasia) is reported to occur at surveillance in 10–20% of cases [55,56].

The previous ER site can be identified by a bland pale area, sometimes with anatomic distortion of the mucosal folds [57]. A standardised imaging protocol for optical assessment of the scar should include high definition white light and narrow band imaging (NBI, Olympus, Inc, Tokyo, Japan) [57]. Optical scar assessment is accurate. A recent multicentre single-blind cross-over trial (*n* = 203) to compare NBI and high definition white light for the assessment of recurrence or residual adenoma at a post-EMR scar reported a negative predictive value (NPV) > 90% (NPV 96% using NBI, NPV 93% using high definition white light) [58]. Use of NBI was not superior to high definition white light (*p* = 0.06) [58]. Expert consensus is that a biopsy is not needed for a bland scar with a uniform pit pattern [57]. Common mimics of recurrence include clip artefact and inflammatory nodules. If an abnormality is suspected, this area should be excised and ablated, as described in a proposed Westmead algorithm for evaluating recurrence [59]. Techniques include cold-snare resection or cold-forceps avulsion with adjuvant snare-tip soft coagulation (CAST), margin ablation and clip closure if any DMI ≥ Type 2 [34,50,60].

## 5. Conclusions

ER is organ-sparing and minimally invasive. It is the recommended primary management strategy for the excision of LNPCPs, supported by high-quality studies. Referral to an expert endoscopist, rather than for surgery, is the standard of care for all patients with an LNPCP. Predicted histopathology underpins the selective resection algorithm and accounts for lesion size, site, granularity, pit pattern and morphology. These resection decision strategies have revolutionised management of LNPCPs. Compared to surgery, they have a lower morbidity and mortality, and are more cost-effective. Unnecessary surgery remains an important issue, and can be overcome by greater awareness of the efficacy and superior risk profiles of ER.

## Figures and Tables

**Figure 1 cancers-15-03805-f001:**
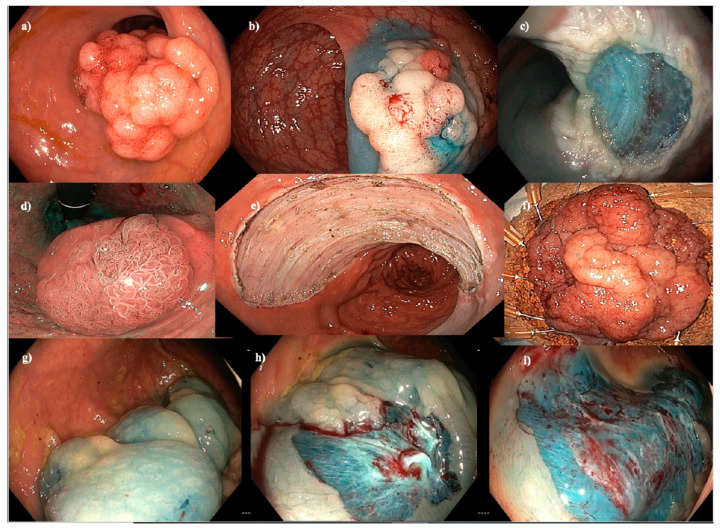
Endoscopic mucosal resection (EMR), endoscopic submucosal dissection (ESD) and cold-snare EMR (C-EMR). (**a**–**c**) EMR of a 40 mm Paris 0-IIa+Is granular hepatic flexure lesion. (**d**–**f**) ESD of a hemi-circumferential 45 mm Paris 0-IIa+Is granular rectal lesion. (**g**–**i**) C-EMR of a 50 mm serrated lesion without dysplasia in a patient with serrated polyposis syndrome.

**Figure 2 cancers-15-03805-f002:**
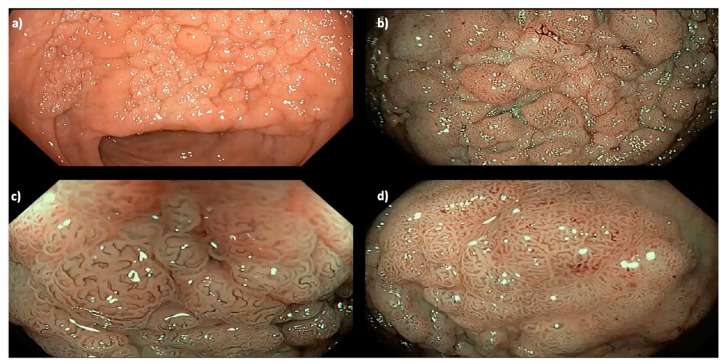
A 35 mm granular Paris 0-IIa LNPCP in the mid-ascending colon, assessed using (**a**) high definition white light, (**b**) narrow band imaging (NBI) and (**c**,**d**) near-focus with NBI, demonstrating a homogenous pit pattern (Kudo pit pattern IV).

**Figure 3 cancers-15-03805-f003:**
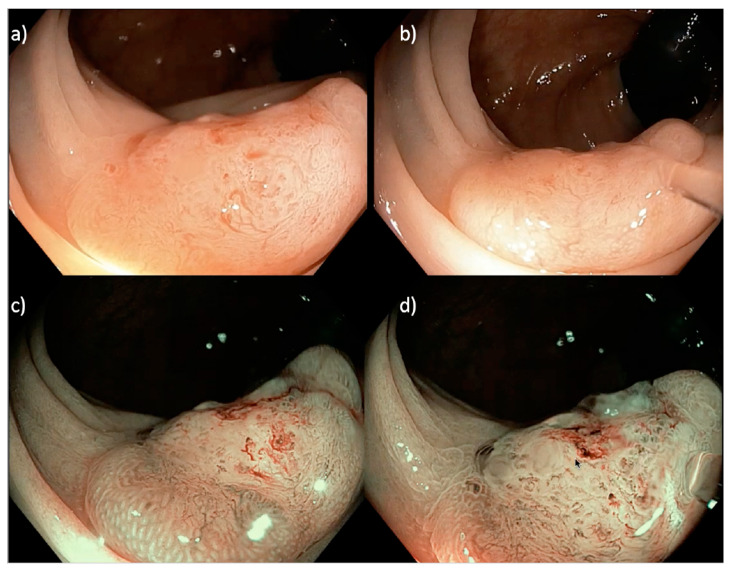
A 20 mm sessile serrated Paris 0-IIa LNPCP in the proximal ascending colon, assessed using (**a**,**b**) near focus and (**c**,**d**) near focus with narrow band imaging (NBI). There is a central well-demarcated area with loss of homogeneity, neovascularization, dilated vessels and a non-structural pit pattern (Kudo pit pattern V_N_), suggestive of a deeply invasive cancer.

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
