# Peer review of "Endoscopic Management of Large Non-Pedunculated Colorectal Polyps"

_cancers, 2023, doi:10.3390/cancers15153805_

Round 1

Reviewer 1 Report

This paper is a very clear REVIEW of endoscopic treatment of LNPCP.

I have only one comment.

Please discuss not only the diagnostic performance of SMIC, but also the long-term prognosis if SMIC is accidentally resected by piecemeal resection. Local recurrence can be controlled by endoscopy, but distant metastasis cannot be controlled by endoscopy and is critical for the patient.

I did not feel that there were any problems with English.

Author Response

Thank you for reviewing this manuscript. 

We agree that discussion regarding post piecemeal endoscopic resection discovery of covert SMIC is common and an important addition to this review. The following has been added: "Until recently, the management of covert SMIC, discovered after piecemeal ER has been challenging. A recent observational study (n=3372) identified 143 (4.2%) cases with covert SMIC post piecemeal resection.48 109 cases underwent surgical resection. 62 (63%) cases had no residual cancer. All cases with residual intramucosal cancer (n=24) could be identified by a R1 histological deep margin. Cases with poor differentiation and/or lymphovascular invasion had a high risk of lymph node metastases (12/33) with a. very low risk without these features (<1%, 0/35). The majority patients with covert SMIC, resected piecemeal have no residual malignancy. The risk of malignancy can be predicted from poor differentiation, lymphovascular invasion and an R1 deep margin."

Reviewer 2 Report

I was glad to review the work of the authors regarding this very interesting review on the Endoscopic management of large non-pedunculated colorectal polyps.

Despite the major advances in colorectal cancer surgery and gastroenterology, there are still numerous unanswered questions regarding the ideal method to remove large non-pedunculated colorectal polyps ≥20mm (LNPCPs).

This is a well-written narrative review. The incorporated images make the study easy to follow.

Major revision:

1) The authors should state the aim of this review clearly in the abstract as well as in the main text.

2) Add at least one table with positive and negative of each other methods that are used for LNPCPs removal

3) ER is organ-sparing and minimally invasive, which enables avoiding wound infections as well as other postoperative complications after open surgery, which cause pain and suffering to patients.  Postoperative complications have been associated with negative economic impact, increased morbidity, extended postoperative hospital stay, readmission, sepsis, and death. I would suggest adding this important information to the discussion section and consider citing the recently published articles

https://pubmed.ncbi.nlm.nih.gov/35371356/

Minor revision

 “In the last few years, technological developments in the surgical field have been rapid and are continuously evolving. One of the most revolutionizing breakthroughs was the introduction of the IoT concept within the surgical practice.”

Add this information in the discussion section and explain the role of IoT in colorectal surgery.

Consider citing the article on the Internet of surgical things

https://pubmed.ncbi.nlm.nih.gov/35746359/

Author Response

Thank you for reviewing this manuscript and providing valuable feedback. We have addressed each comment below:

1) Aims now included in abstract and introduction. "This review aimed to provide a comprehensive, critical and objective analysis of ER techniques.

2) While the addition of a table could be helpful in comparing technical differences between modalities (eg a table comparing technique differences for hot vs cold EMR), we do not believe a table comparing advantages and disadvantages of each resection modality would be helpful given each has a unique place within the endoscopic resection algorithm.

3) Thank you for this suggestion. We agree, that the addition of the following is important. "The rectum should be regarded as a complex high-risk site, with distinct challenges compared to the colon. This is not due to its technical limitations but due to its increased risk of covert SMIC.26, 40 Furthermore, the consequences of failed endoscopic cure include consideration of the most hazardous and complicated forms of colorectal surgery including permanent ostomy formation 41. Patients with rectal lesions removed by a low or ultra-low anterior resection have an increased risk of: incontinence (12%) 42, sexual dysfunction (20-46%) 43 and a 10-20% risk of permanent stoma 44, 45. Low anterior resection has a 30 day morbidity and mortality of 25% and 6% respectively 46. Postoperative complications have been associated with negative economic impact, increased morbidity, extended postoperative hospital stay, readmission, sepsis, and death. ER is organ-sparing and minimally invasive, which enables avoiding wound infections as well as other postoperative complications after open surgery, which cause pain and suffering to patients.47 "